# Fiber Tensioning Systems in a Robotized Winding Procedure for Composite Materials Building Processes

**Mădălin Petru Sbanca *** , **Răzvan Gabriel Boboc** and **Gheorghe Leonte Mogan**

Department of Automotive and Transport Engineering, Transilvania University of Brașov,
RO-500036 Brasov, Romania
* Correspondence: madalinsbanca@gmail.com

**Abstract:** The winding process of carbon fiber is not a new concept, but until now only a few studies have been made in carbon fiber tensioning and winding on complex shapes. The main purpose of the article is to analyze the possibility to conduct the carbon fiber winding using two industrial robots connected as master slave, one to realize the winding and the other to hold and rotate the mandrel, using an own design of an automatic fiber tensioning tool. To have control of the system a force transducer was implemented. The measured force was used to make trajectory adjustments for the robots, having a predefined trajectory. The results obtained from the experimental tests showed that the winding process with two robots and a tensioning system can realize fiber windings with variable tension in molds with complex configurations (3D) of the winding paths. In this way, compact structures of the constantly tensioned fiber bundle can be obtained, according to product requirements.

**Keywords:** fiber tensioning; winding process; composite materials; industrial robots

## 1. Introduction

Composite materials and structures are now being exploited in a wide range of applications from the automotive to the medical industry [1–3]. There are some important advantages, such as high strength, excellent fatigue properties, and low weight. Additionally, they do not corrode, can tailor laminate to loads, and can enable very significant part count reduction.

Composite materials are manufactured from at least two components—fiber and resin. Advanced composite materials, widely used for the aerospace industry, have a specific structure and reinforcement that can be made out of carbon fiber [4], fiberglass [5], or Kevlar [6].

The fibers can be made of different materials, but the most used is carbon fiber. Fibers are bundled together to make yarns. The production of carbon fiber, in the latest years, increases significantly [7], enabling the development of the industry.

Typically, there are two types of carbon fiber: pre-impregnated fiber, meaning an additional process is not needed during its manipulation, or dry fiber, for when it is necessary to consider a resin bath during the manipulation.

There are some steps to be followed to obtain the pre-impregnated carbon fiber. The carbon fiber is unrolled from the tow and placed in a resin bath, directed by a rolls system. From the resin bath, it is placed in a high-speed oven to dry back and then rolled to a tow. The impregnation of the fiber during the manipulation is realized in a similar way, except that the oven is missing in this step, but it is placed after the complete part is ready.

The winding process is a general procedure, used in several areas. The winding process can be done differently depending on the product and technological evolution. According to [8], compared to conventional winding techniques, cable winding includes

fewer manufacturing steps and is therefore likely to be better suited for automated production. Automation of the cable winding production step is a crucial task to lower the manufacturing costs of these machines.

A robot cell offline simulation was posed in [8] and used during the development of a functional automated cable winding production cell to validate its function. It is presented as a robotic cell with ABB Robots used for copper fiber winding around the stator. The process contains a feeding component [9] to control the fiber quantity, but also to keep the fiber tension constant. In the actual study, in testing the concept phase a Robot Studio simulation was also created, where all the components from the system are placed. Another strategy is to use MATLAB to simulate the trajectory of the robot [10]. In ref. [10], some simulation tests were presented, which are also performed in the actual research, concerning:

- Online trajectory corrections, using MATLAB in [10], in the actual study using Robot-Studio and a self-developed platform in C#;
- Collision avoidance simulations;
- Importing all real components from the environment in the simulation.

MATLAB (R2018b) software is used because the winding die is a cylinder shape, and the mathematical model was available to be calculated. In the actual study, the identification of the mathematical model is complicated to identify because:

- The robot is making the winding following a complex trajectory;
- The influences are from the robot winding but also the carbon fiber source role.

The advantages of winding automatization can be observed in [8]. For the presented process, the costs were reduced from 1.8 to 0.2 million euro. However, the costs are not the only important benefits. Better control of the system is also important, meaning higher quality and accuracy. For this, in [11] some characteristics were studied in more detail, such as fiber tensioning or friction with the deposition head.

In ref. [12], the authors analyze the automatization opportunity of the winding process using carbon fiber processed by a robotic system. It was assumed that the tape tension is as constant as possible, and the deposition head and the robot arms should move on collision-free trajectories. The system also included components such as a feeding device made out of the main frame, a roving guide system, a roving tensioner, and a deposition system. It is important to mention that all of these components are placed, taking into consideration the carbon fiber flow, before the deposition head and before the winding die. The study proves that there is an influence of speed, more visible in the curves. From the beginning, the author considered the fiber tension constant. This is not difficult to obtain for a simple winding process [12], but for a complex die, where also a robot with a complex trajectory is involved, it is impossible to do it without a complex tensioning device. In the robotic winding processes as in the patent [13], the carbon fiber place must be considered. In [12], it is placed directly on the robot arm, next to the deposition head. This solution does not consider dry carbon fibers. Moreover, it is not considering the necessary space and also to change the fiber tow it is necessary to stop the process, meaning that the human has to interact with the robot. In addition, one of the missing aspects of this solution is the tensioning option, but the method of implementing the trajectory, by using the virtual model of the winding die from where the nominal trajectory of the robot is generated. Parallel to this, in [14] a specialized software made in MATLAB is proposed to generate the trajectory for a single robot system. In ref. [15], winding using a single robot is proposed, using a tensioning device but a standard model. In an extension, the work can be considered as using a multi-robot winding system, because it proposed a mobile winding die for a simple trajectory.

According to [15] the programming effort of teaching robots winding is currently high due to missing commercial path generation software, standard ones, not generated using the CAD drawing line, as in [13]. The effort for teaching two cooperative robots is significantly higher than for one single robot. Collaborative robots tend to be used in more

sensitive areas, such as medicine, where using other robot assistant systems, such as force control, can be used with a high accuracy [16]. The disadvantages of these systems are the speed and the environmental conditions.

The necessity of using multiple robot cells comes from the complexity of the winding die. There are three categories of winding die:

- Fix die;
- Single-axis winding die;
- Multiple axis winding die.

For the first type, a single robot or actuator system is enough to proceed with the winding. For the second category, in most cases, a robot and a mobile winding die are used to complete the robot's reachability points [15]. For the last option with multiple-axis winding dies [17], it is recommended to use a multiple-robot system. The advantages are related to the flexibility of the system, speed, and accuracy. In this study, it is also considered a multi-axis winding die with a multi-robot system.

For the manufacturing of new materials made with carbon fiber, there are several modeling methods [18]. For the moment, the modeling methods are not in a high process of automatization, even the composite materials are a necessity of every major company in the mentioned industries. The reasons for this lack are connected to the high price of carbon fiber and its modeling methods [19]. These methods are not yet very stable and suitable for automatization.

In this study, a solution for the modeling method of carbon fiber is proposed. Furthermore, the purpose of the study is to identify automatization methods for carbon fiber modeling that are fast, low cost, and more efficient. We also want to obtain an increased quality of the resulting composite materials. Two of the most important modeling methods of carbon fiber are considered: the placement method and especially the winding method. From the previous studies [20–25], it was observed that in the winding process the most important parameters are the trajectory, speeds, and especially tension of the fiber during the process.

The main objective of this research is to obtain a complete system, hardware and software, which can build composite materials using the winding process taking into consideration the most important composite structure characteristics. For the automatization of the winding process, we considered other continuously developing areas for automatization, respectively, the robotization of the processes.

The research proposed an integrated winding system for carbon fiber using two industrial robots and tensioning systems in order to obtain a fast process for finite composite parts. Thus, we propose a new production method for composite materials made from carbon fiber using the winding process with a high potential on both a high volume of parts and unique objects production.

## 2. Variable Tensioning Device

### 2.1. Actual Methods

Besides the mechanical implementation of the adjustable tensioning system, it is important to take into consideration also the software part, which is including the measurement systems, the drive systems, and the control systems. In ref. [4] or [26] it is considered a typical control system, PID, with manual adjustments. The designed control system consists, in [4], of a magnetic brake, a servo motor, a PID control unit, a load cell, and a data converter. The tension of the carbon fiber was measured by a load cell and compared to the preset value to keep the tension of the carbon fiber in a predefined certain range. The manual adjustments are made in a system where the mathematical model is impossible to determinate or during the process the mathematical model is changing. According to the methodology of the winding process, the pretension load on the fiber during the winding process must be constant on an acceptable value. The system presented in [4] or [27] is much used for reducing the complexity of the winding die. Furthermore [4], in [26] a system to brake the fiber tow and the winding die is also used. The difference is

that the winding die is also a cylindrical form so the winding die and the fiber tow can be synchronized according to the brake. On this type of winding, the mathematical model can be also established with accuracy. Therefore, the control system model is known.

In ref. [26], based on the mathematical model of the system, a control system is proposed with calculated and simulated formula. On the software level, the components of the control system were taken into consideration as presented in Figure 1. In the actual study, a control system with two feedback loops is proposed. One loop is for the tensioning servomotor positioning, using the position feedback, and one loop is for the tensioning itself using the force feedback. There are also models, such as [28] or the patent from [29], which try to avoid a control system on the software level and use a rolls system with springs. The results from this type of system are acceptable according to the tests, but a big disadvantage is size. The necessary space depends on the requested accuracy. Taking into consideration the technological advance in the calculation process, a complete mechanical system is not a solution that can be generally accepted.

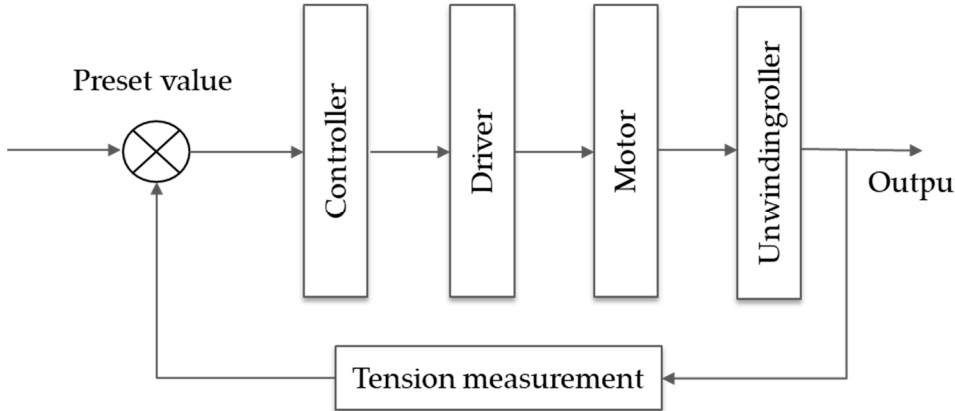

**Figure 1.** Control system of the tensioning.

Study [26] proposes a dynamically controlled system. This type of system is used to limit the tolerances of the system and the compensation to be made faster, as presented also in the actual study. In ref. [27], it is pointed out that in a roll-based system the strength of the carbon fiber is reduced considerably, to 40% depending on the pre-tensioning unit parameters. However, the damage to the carbon fiber increase because of the increased friction of the fiber with the winding die and also because of the pretension force on the carbon fiber in a constant pre-tensioning system.

### 2.2. In system Tensioning Device

A model for the actual study is presented in [11], where a winding model of copper fiber with the tensioning system using rolls in the deposition head creates friction with the fiber, therefore tension between the winding die and the fiber. Starting from [8,11], it was clear that errors in the winding procedure, such as slipping on the cable while feeding and slipping in the tool feeding mechanism, must be quickly detected and avoided if possible. The alternative is proposed in [30], where a new monitoring system integrated inside the FRP construction can measure material wear and fatigue, as well as material load. Such recording and evaluation systems are necessary to precisely locate the critical structural changes inside the composite to reduce production and maintenance costs. The disadvantage is the initial costs for the sensor.

The tensioning tool design should be simple, robust, easy to adjust and maintain, and not unnecessarily expensive. It should be possible to mount on the tool and be able to communicate with an industrial robot. Using step motors to control the feeding wheel and the guiding system provided high control of these functions at a reasonable cost. Hereby synchronized feeding of the cable through the stator, using both feeder tools to push and pull the cable through the same slot hole, could be achieved. In ref. [11] an error

handling system was implemented to react to possible differences from nominal during the process progress. In the actual study, we preferred to use a control procedure base on the feedback coming from the sensors in the fastest time possible (Figure 2). In this option, every difference will be handled with a mathematical calculation and the result will be compensating directly. Supervised error handler functions in the robot programming were used to detect and react to errors in the tool function and during winding. Different error functions were activated and reset depending on the current winding task.

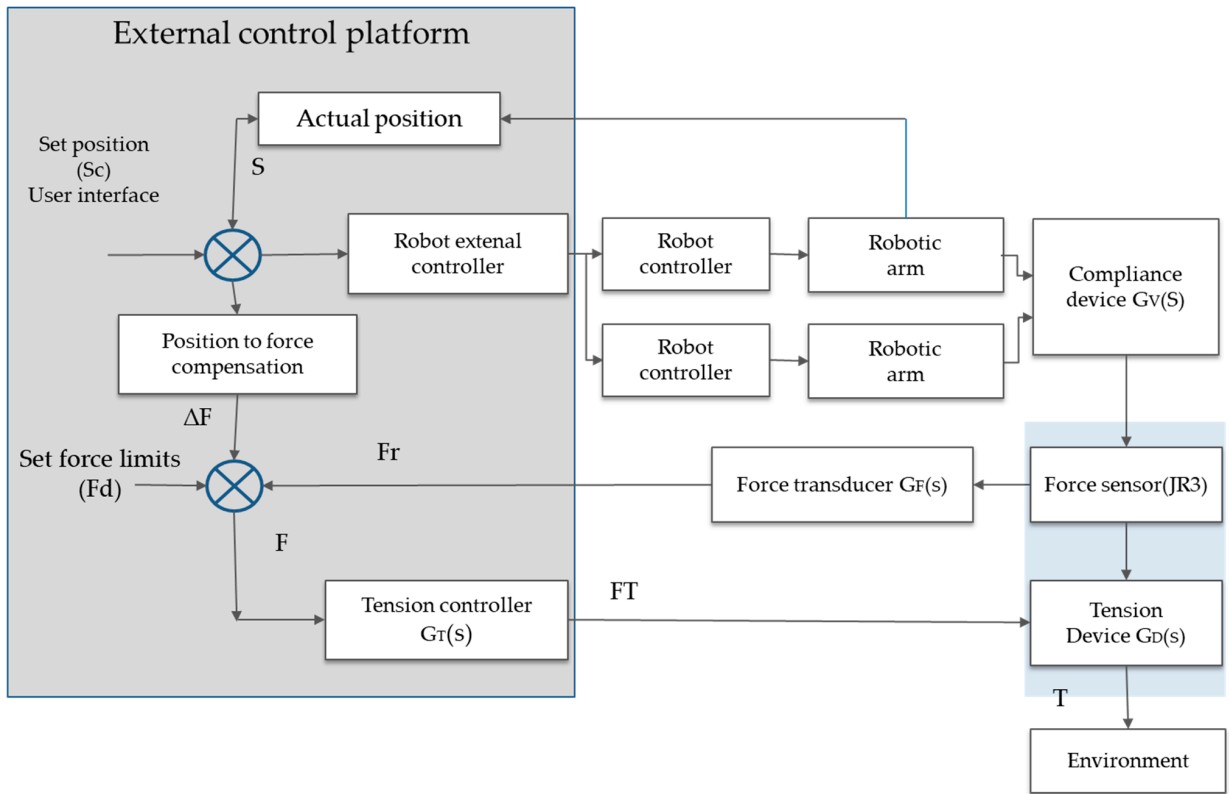

**Figure 2.** Software and hardware control system of the tensioning.

### 2.2.1. Nominal Parameters Identification

The main component of the tensioning system is the tensioning motor. This is transforming the rotation into a linear position and then in contact with the fiber in tension on the carbon fiber. The motor must be positioned with high accuracy. Therefore, a control system is needed.

The first step in the control system design is the identification of the mathematical transfer function, which indicates the most realistic relation between applied tension and the angular motor speed. So, the process input is the tension force, and the output is the angular position of the motor shaft. For simplicity reasons and to ignore the transitional electric model, the motor dynamic is considered a level 1 delay element, presenting an amplification factor and a time delay constant:

$$Gm(s) = \frac{V(s)}{U(s)} = \frac{K_m}{sT_m + 1} \tag{1}$$

It is important to identify the parameters which define the motor dynamic. The amplification factor is defined as:

$$K_m = \frac{Ouput\ stationary\ value}{Process\ input\ value} \tag{2}$$

The input parameter is the motor tension, and the output is the angular speed. Therefore, the amplification factor can be described as the ratio between the stationary value of angle speed and the applied tension on the motor rotor. These values can be identified from the motor description.

$$K_m = \frac{Angle\ speed\ for\ free\ run(rpm)}{Power\ tension\ (V)} = \frac{6700}{10} = 670 \tag{3}$$

The amplification factor was also calculated by the simulation to confirm the value obtained from the motor description. The values obtained at a 4700 rpm speed with 7 V power tension were measured and the result is similar (Figure 3).

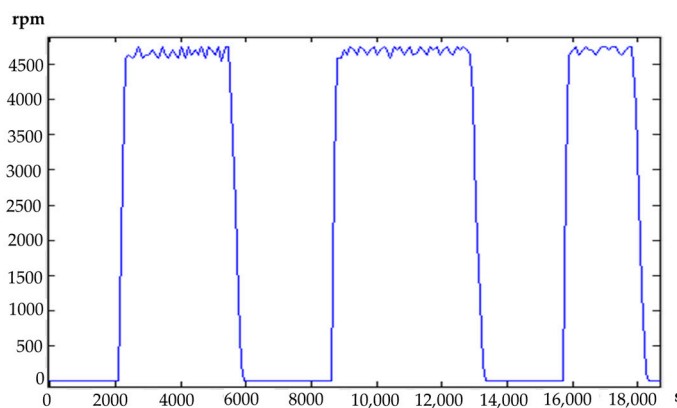
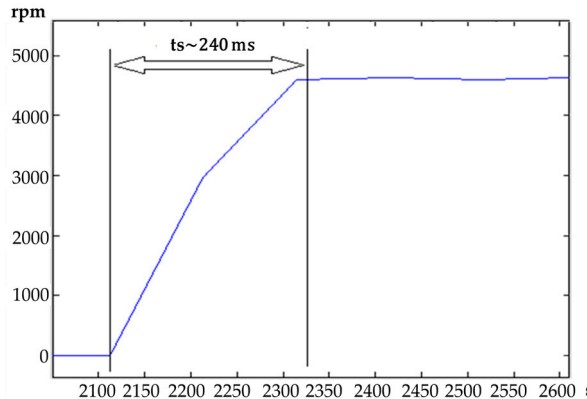

**Figure 3.** Motor amplification factor simulation.

Settling time is very small because is a fast process. The time value was established at 200 ms after the simulation. Because of the transitional model results:

$$T_m = \frac{ts}{3} \sim 80\ ms \tag{4}$$

$T_m$ time constant is very small, but cannot be avoided, therefore we establish $T_m$ = 0.08 s.

The transfer function for the complete system for the angle position will be:

$$Gm(s) = \frac{\theta(s)}{U(s)} = \frac{K_m}{s(sT_m + 1)} \tag{5}$$

The translation from angle speed to angle position is made using an integrator factor:

$$\theta(s) = \frac{V(s)}{s} \tag{6}$$

Considering $V(s) = s * \theta(s)$, therefore $v(t) = \frac{d\theta(t)}{dt}$.

For the control system design, there are two performance criteria, the settling time—smaller than 500 ms and override not bigger than 5%, to avoid vibration on the motor.

Proportional Control System

The first proposal is a proportional (P) control system.

If $T_m$ from Equation (5) is not considered, the initial transfer function would be:

$$Gm(s) = \frac{K_m}{s} \tag{7}$$

Adjusting function Gk(s) = KR, the transfer function for the system with open loop is:

$$Gd(s) = Gm * GR = \frac{K_R * K_m}{s} \tag{8}$$

The system transfer function in close loop will be a level 1 element:

$$G_0(s) = \frac{Gd(s)}{1 + Gd(s)} = \frac{K_R * K_m}{s + K_r * K_m} = \frac{1}{\frac{s}{K_r * K_m} + 1} = \frac{1}{sT0 + 1} \tag{9}$$

$$T0 = \frac{1}{K_R * K_m}, \tag{10}$$

ts = 3*T0, then:

$$k_R = \frac{1}{T0 * K_m} = \frac{3}{ts * K_m} \tag{11}$$

The time constant, $T_m$, cannot be avoided, even if it is small, because it is increasing the override value:

$$Gd(s) = Gm * GR = \frac{K_R * K_m}{s(sT_m + 1)} \tag{12}$$

$$G_0(s) = \frac{K_R * K_m}{s(sT_m + 1) + K_R * K_m} = \frac{K_R * \frac{K_m}{T_m}}{s^2 + \frac{1}{T_m}s + \frac{K_R * K_m}{T_m}} \tag{13}$$

$$G_0(s) = \frac{\omega n^2}{s^2 + 2\,\zeta\,\omega n + \omega n^2} = \frac{K_R * \frac{K_m}{T_m}}{s^2 + \frac{1}{T_m}s + \frac{K_R * K_m}{T_m}} \tag{14}$$

The resulting transfer function is a level 2 element. The level 2 system allows the existence of three operating functions system: underdamped, critically damped, and overdamped, depending on the value of the damping factor, $\zeta$.

The underdamped system, the settling time is:

$$ts = \frac{4}{\zeta\,\omega n} = 8T_m = 8 * 80 = 640 \text{ ms} > 500 \text{ ms}$$

Not depending on the control system parameters, but only on the settling time, which means there is no control on the settling time, being a constant, bigger than the set value.

On the underdamped system, the override can appear, expressed by:

$$Mv = \exp(\frac{-\pi\zeta}{\sqrt{1 - \zeta^2}}) \tag{15}$$

Override is a descending function. To minimize the override, the damping factor must be increased.

$$\zeta = \frac{1}{2\sqrt{k_R * k_M * T_M}} \tag{16}$$

Increasing the damping factor means the amplification factor of the control system decreases. To avoid override, we can request a critically damped system, which is the nominal system with 0 overrides.

In critically damped system $\zeta = 1$, therefore:

$$2\sqrt{k_R * k_M * T_M} = 1 \tag{17}$$

$$k_R = \frac{1}{2\sqrt{k_M * T_M}} \tag{18}$$

In this system, the values established for settling time and override are not the same.

Even if the override is 0, the formula for settling time is difficult to establish analytically, but also the control system constant is already fix, and it cannot be modified for an

eventually change during the settling time. Having only one parameter for the control system, cannot adjust two parameters, settling time and override, ts and Mv.

A better option is by adding a derivative component to the control system.

Proportional Derivative Control System

According to the time constant compensation rule from the process to the control system, a control system with the derivative component was selected. The mathematical model is already an integrator in a direct way, which ensures null stationary error in close loop. Therefore, a control system with no integrative action must be selected. The proposed control system is based on a proportional derivative rule. There is added a filter to fulfill the achievement conditions. In the end, the control system will be a PDT1.

For the numeric implementation, the filter would not be necessary, but for a real implementation, it is necessary.

$$G_R(s) = \frac{k_R(sTd+1)}{sTf+1} \tag{19}$$

$$Gd(s) = G_R(s) * G_M(s) = \frac{k_R * k_M * (sTd+1)}{s(sTf+1)(sT_m+1)} = \frac{k_R * k_M}{s(sTf+1)} \tag{20}$$

It is selected $T_d = T_m$ to compensate the time constant from the process:

$$G_0(s) = \frac{Gd(s)}{1+Gd(s)} = \frac{k_R * k_M}{s^2Tf + s + k_R * k_M} = \frac{\frac{k_R*k_M}{Tf}}{s^2 + \frac{1}{Tf}s + \frac{k_R*k_M}{Tf}} \tag{21}$$

For the control system design, the module criteria that fulfill an override of Mv = 4, 3% < 5% was used.

The transfer function is in a close loop according to the modulo criteria, to fulfill $\zeta = \frac{1}{\sqrt{2}}$ is:

$$G_0(s) = \frac{\omega_0{}^2}{s^2 + \sqrt{2}\omega_0 s + \omega_0{}^2} \tag{22}$$

$$\omega_0{}^2 = \frac{k_R * k_M}{Tf} \tag{23}$$

$$\frac{1}{Tf} = \sqrt{2}\omega_0 \tag{24}$$

Therefore,

$$ts = \frac{4}{\zeta\omega_0} = \frac{8}{2\zeta\omega_0} = 8Tf => Tf = \frac{ts}{8} \tag{25}$$

$$k_R = \frac{\omega_0{}^2 Tf}{k_M} = \frac{\frac{1}{2Tf^2}Tf}{k_M} = \frac{1}{2Tfk_M} = \frac{8}{2tsk_M} = \frac{4}{tsk_M} \tag{26}$$

The PDT1 control system parameters are:

$$\begin{cases} k_R = \frac{4}{tsk_M} \\ Tf = \frac{ts}{8} \\ Td = Tm \end{cases} \tag{27}$$

The override the set to 0.5 s:

$$\begin{cases} ts = 0.5 \text{ s} \\ k_M = \frac{670*2\pi}{60}\frac{rad}{s} \\ Tm = 0.08 \text{ s} \end{cases} \tag{28}$$

$$\begin{cases} k_R = 0.114 \\ Tf = 0.0625 \\ Td = 0.08 \end{cases} \tag{29}$$

The transfer function of the control system is:

$$G_R(s) = \frac{0.114(0.08s + 1)}{0.0625s + 1} \tag{30}$$

To realize the real implementation base on identified close loop transfer function, the control system discretization was made, using the substitution method with $s = \frac{1-z^{-1}}{Te}$, corresponding to the numerical integration of the backward rectangular rule.

$$G_R(z) = G_R(s), \ cu \ s = \frac{1 - z^{-1}}{Te} \tag{31}$$

$$G_R(s) = \frac{k_R(sTd + 1)}{sTf + 1}, \ cu \ s = \frac{1 - z^{-1}}{Te} \tag{32}$$

$$G_R(z) = \frac{k_R\left(\frac{1-z^{-1}}{Te}Td + 1\right)}{\frac{1-z^{-1}}{Te}Tf + 1} = \frac{k_R\left(\frac{Td-Te}{Td} - z^{-1}\frac{Td}{Te}\right)}{\frac{Tf+Te}{Te} - z^{-1}\frac{Tf}{Te}} = \frac{k_R\left(\frac{Td+Te}{Tf+Te} - z^{-1}\frac{Td}{Tf+Te}\right)}{1 - z^{-1}\frac{Tf}{Tf+Te}} = \frac{b_0 + b_1 * z^{-1}}{1 + a_1 z^{-1}} \tag{33}$$

Considering:

$$G(z^{-1}) = \frac{U(z^{-1})}{E(z^{-1})} = \frac{b_0 + b_1 * z^{-1}}{1 + a_1 z^{-1}} \tag{34}$$

$$U(z^{-1})(1 + a_1 z^{-1}) = E(z^{-1})(b_0 + b_1 * z^{-1}) \tag{35}$$

$$U(z^{-1}) + a_1 z^{-1} U(z^{-1}) = b_0 E(z^{-1}) + b_1 * z^{-1} E(z^{-1}) \tag{36}$$

and

$$G_R(s) = \frac{0.114(0.08s + 1)}{0.0625s + 1} \tag{37}$$

$$Z(u[k - n]) = z^{-n} U(Z^{-1}) \tag{38}$$

Then,

$$u[k] + a_1 u[k - 1] = b_0 e[k] + b_1 * e[k - 1] \tag{39}$$

Therefore,

$$u[k] = -a_1 u[k - 1] + b_0 e[k] + b_1 * e[k - 1] \tag{40}$$

and

$$\begin{cases} a_1 = -\frac{Tf}{Tf+Te} = -0.7576 \\ b_1 = -\frac{Td*k_R}{Tf+Te} = -0.1106 \\ b_0 = k_R\frac{Td+Te}{Tf+Te} = 0.1382 \end{cases} \tag{41}$$

Making the substitution, a mathematical equation results, which is implemented in a microprocessor.

$$u[k] = -0.7576 * u[k - 1] + 0.1382 * e[k] - 0.1106 * e[k - 1] \tag{42}$$

The control system and the mathematical equation were tested first in a simulation using MATLAB and Simulink. In Figure 4, the simulation of the control system in two situations is presented. The first one is the theoretical one, where the power tension is not limited. In this case, the control system fulfills completely the requested performance criteria of Mv = 4.3% and ts < 500 ms. The second situation considers the saturation of the input element, power tension being limited to 7 volts. In this case, the settling time cannot

be fulfilled as requested. From the tests presented in Figure 4, it results that, for one rotation (360 degrees), the input is in saturation for 700 ms, until the angle is 300 degrees, the control system is activated only in the last moment to brake the motor without oscillations.

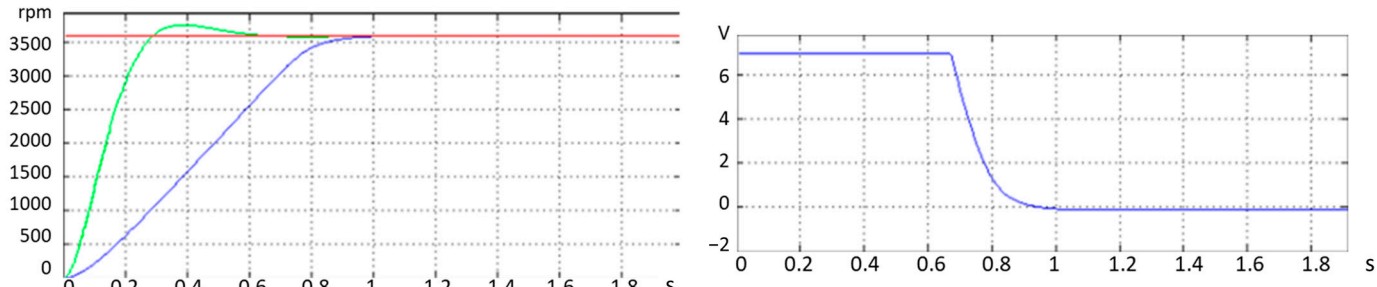

**Figure 4.** Simulation of the control system.

The settling time is still in the range, $t_s$ = 800 ms, which was confirmed also after the real test. The motor was in position (one rotation) in time, with no oscillation, with a small override of a few degrees.

The reference is for 10 rotations from the 0 point. The control system that was tested was made out of a range of rectangle impulses, setting to the motor ten rotations or 0 rotations alternatively. The power tension was limited to 7 Volts. The simulation results of the control system are presented in Figure 5.

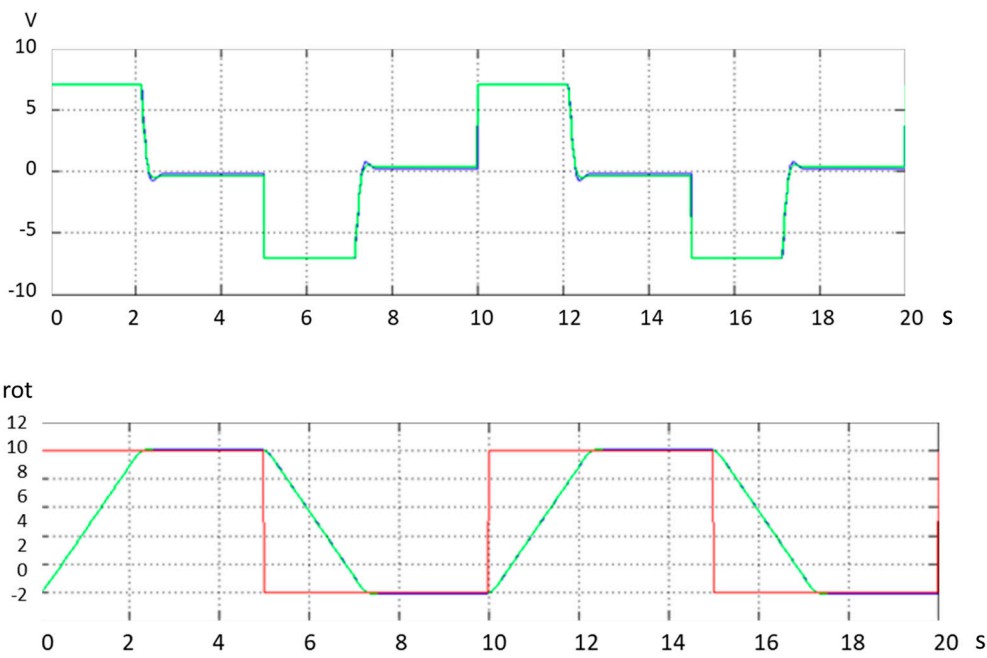

**Figure 5.** Control system simulation for limit rotations.

After the control system starts, there is still a small override, but this is not influencing the general results, only the influence of the motor inertia. After the simulation, it is proved that the control system was selected correctly, and it can be applied to the motor. Without saturation, the requested settling time of 0.5 s could be achieved, but in this case 150 V for motor starting would be necessary, as simulated in Figure 6.

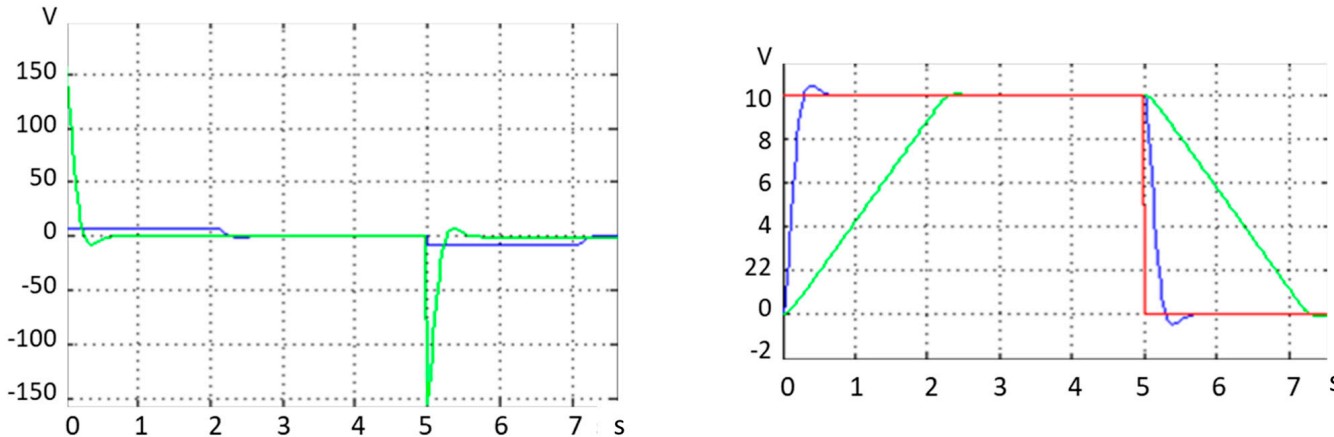

**Figure 6.** Tension inputs for settling time achieve.

With a saturation effect on 7 V tension, the settling time according to the simulation is approximately 2.2 s. In Figure 7, the overview and in Figure 8 single step of the real measurements from the motor are presented. These are similar to the simulation and fulfill precisely the position requirements.

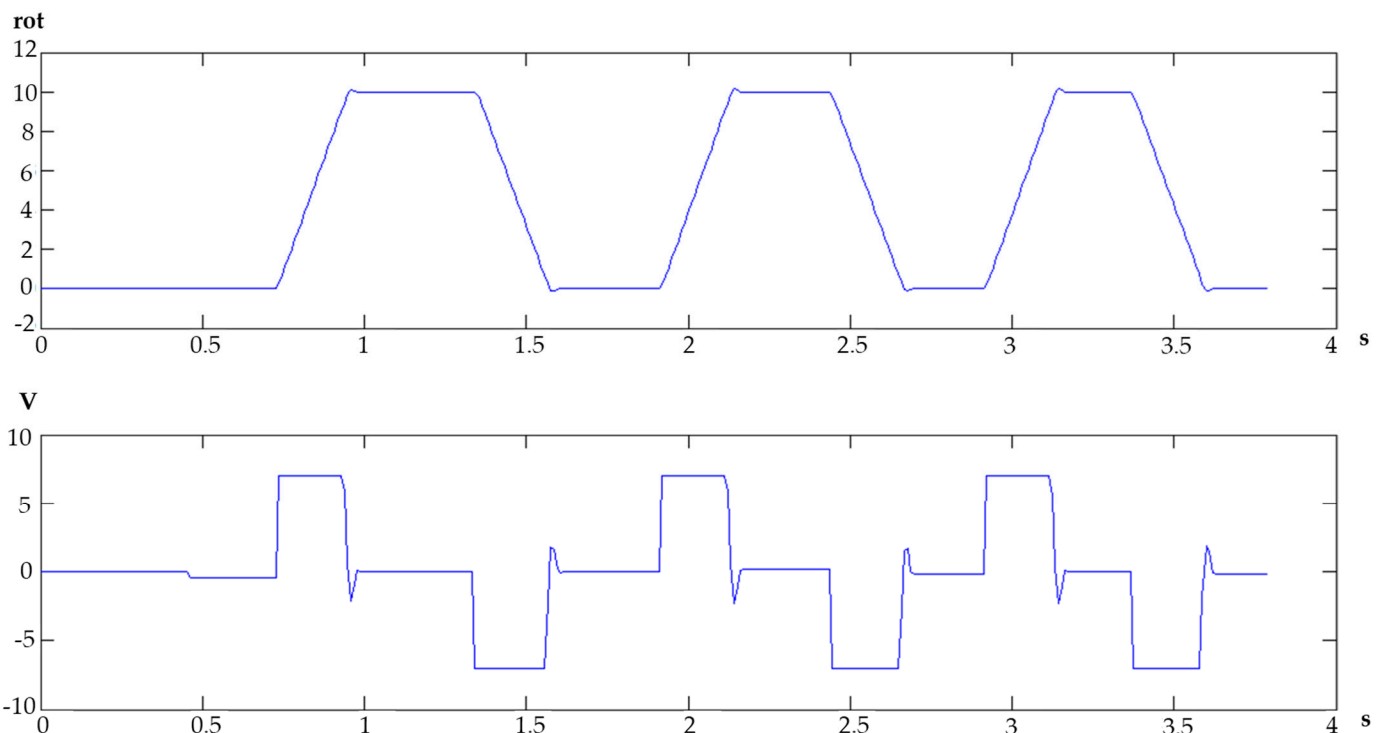

**Figure 7.** Real motor positioning.

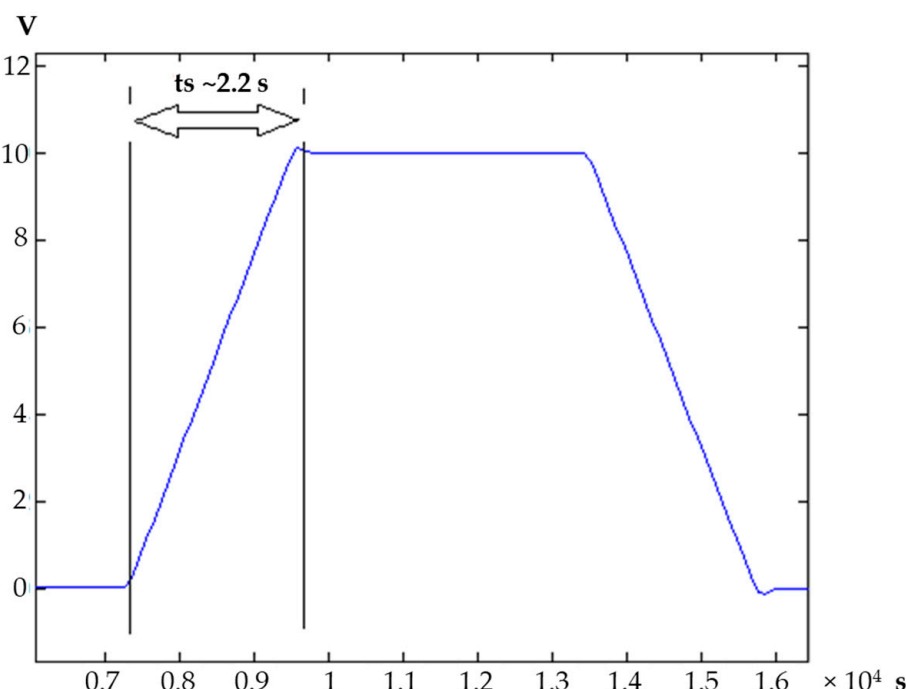

**Figure 8.** Single-step motor real measurements.

### 2.2.2. Automatic Tensioning System

The standard tensioning models were not suitable for our goal, therefore another type of mechanical implementation was proposed (Figure 9a), according to the patent in publishing.

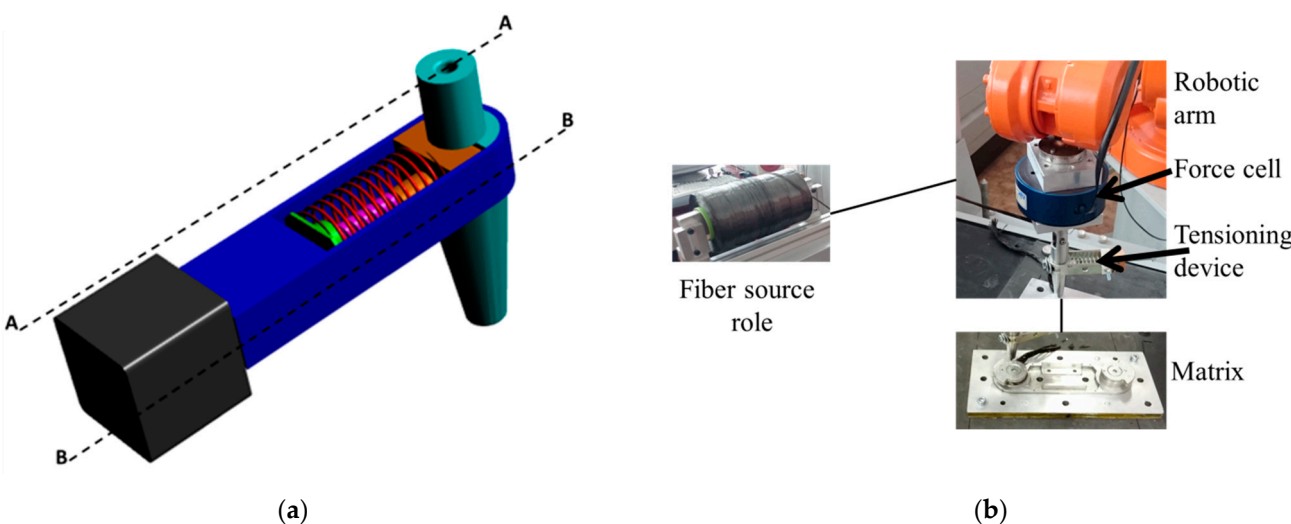

(**a**)　　　　　　　　　　　　　　　　　　　　　　(**b**)

**Figure 9.** (**a**) Automatic fiber tensioning tool; (**b**) Real system implementation.

The system includes a motor pressing on a spring and then on the fiber, creating tension between the mandrel and the deposition head (Figure 9b). The big advantage of this model is that the tensioning point is very close to the winding process on the winding die. The positions and process before the winding head are not influencing the process itself (Figure 10).

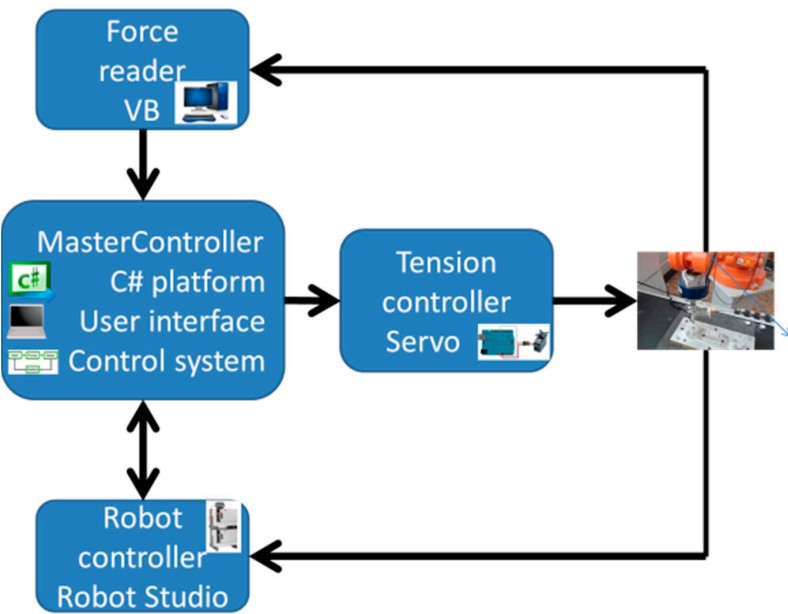

**Figure 10.** System diagram.

For this system (Figure 10), we needed a mathematical model to know how to control it and how to transform the set force in the fiber tension into the motor position. The first proposal is:

$$Gm(s) = \frac{\theta(s)}{U(s)} = \frac{K_m}{s(sT_m + 1)} \qquad (43)$$

To be applicable in a controller, the formula is adjusted to:

$$u[k] = -0.7576 * u[k-1] + 0.1382 * e[k] - 0.1106 * e[k-1] \qquad (44)$$

This result was applied to an Arduino microcontroller for the position adjustment. The position is sent from the master platform in value of the number of turns of the motor.

The force was measured without using the tensioning device and the results can be seen in Figure 11.

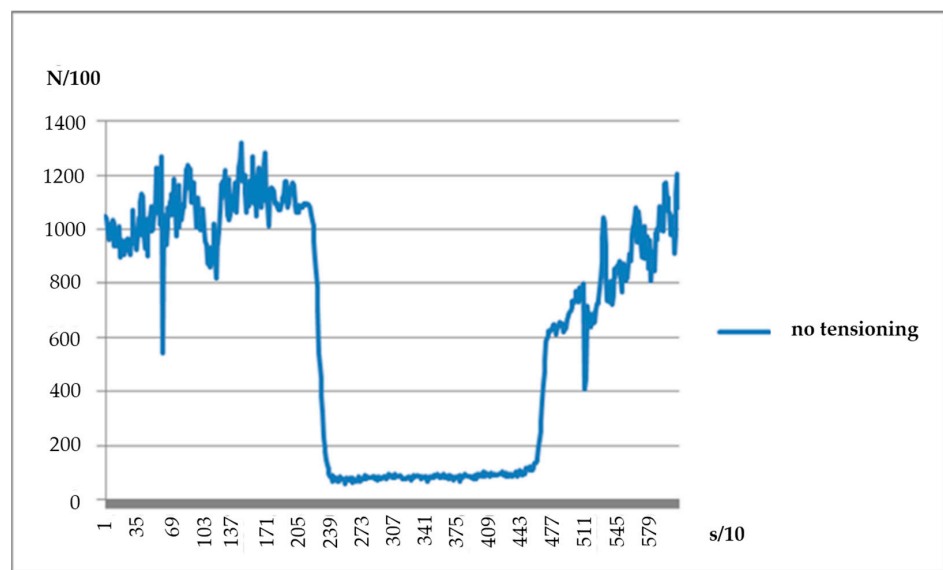

**Figure 11.** Tension force measurement without force control.

The force differences are from the perturbation imposed by the fiber source role, which is placed on a side, and it has to be unrolled by the robot movement. This unrolling process is made using force, which in the end is applied to the winded fiber. When the roll is already unrolled, the winding force is too weak.

The force cell used is giving feedback on all three axes (X,Y,Z). Because the movement is not straight to one of the directions, in the following equations the force vector was calculated based on the force feedback on all three directions as a square root from the forces on each direction:

$$messforce = \sqrt{force\ X^2 + force\ Y^2 + force\ Z^2} \tag{45}$$

$$forta\ brut = \sqrt{forta\ X^2 + forta\ Y^2 + forta\ Z^2} = \sqrt{Xudp^2 + Yudp^2 + Zudp^2} \tag{46}$$

Considering position 0 with the force at the maximum applied to the fiber, the formula which converts the measured force to the position is:

$$PozVersion1 = \frac{actual\ force * 10}{reference\ force} = \frac{messforce * 10}{avgforce} \tag{47}$$

This formula uses a reference force which was selected to be the maximum force that can appear when the position is at maximum and the force is coming from the fiber source role.

The conversion force to the position was considered at the beginning to be linear and proportional and, because of the inverse effect on the real system, the force and position were automatically adjusted by the process.

### 3. Simulations and Study Results

*3.1. Mathematical Model*

Three tests were conducted in the experiments. The first one uses the formula with a reference of the motor having at 0 rotations, the reference force 500 [N/100], (Figure 12—blue line). The second test was made with the reference force modified to 1000 [N/100], (Figure 12—red line). During the tests, it was observed that the used spring is not strong enough to press on the fiber with the necessary force to be stronger in a normal state than the one coming from the fiber role perturbation. The last tests, (Figure 12—grey line) were conducted with a changed spring, stronger, but keeping the modified reference. From this, the high force is also coming at the beginning. It is combining perturbation, from the source role, with the tensioning force. To have a bigger amplifier, the reference was increased to 1200 [N/100], which is closer to the source role average force to be applied in the position, and also an amplifier factor was added.

$$PozVersion2 = \frac{messforce * 10}{reference\ force} + 5\left(\frac{messforce - 1200}{avgforce}\right) \tag{48}$$

After the formula was changed, some improvements can be seen, but in some points, we can still see a big drop in the tension, which will influence the properties of the final product.

An additional amplifier was added in order to make the relation between force and position more stable for the experimental tests:

$$PozVersion3 = \frac{messforce * 10}{reference\ force} + 10\left(\frac{messforce - 1200}{1200}\right) \tag{49}$$

Even if the amplifier value was changed, it can still be seen drop of the tension, and even worst high fluctuation on the force value.

$$pozitie\ Varianta\ 4 = \frac{actual\ force * 10}{reference\ force} + 12\left(\frac{actual\ force - 1200}{1200}\right) = \frac{forta\ brut * 10}{forta\ medie} + 12\left(\frac{forta\ brut - 1200}{1200}\right) \quad (50)$$

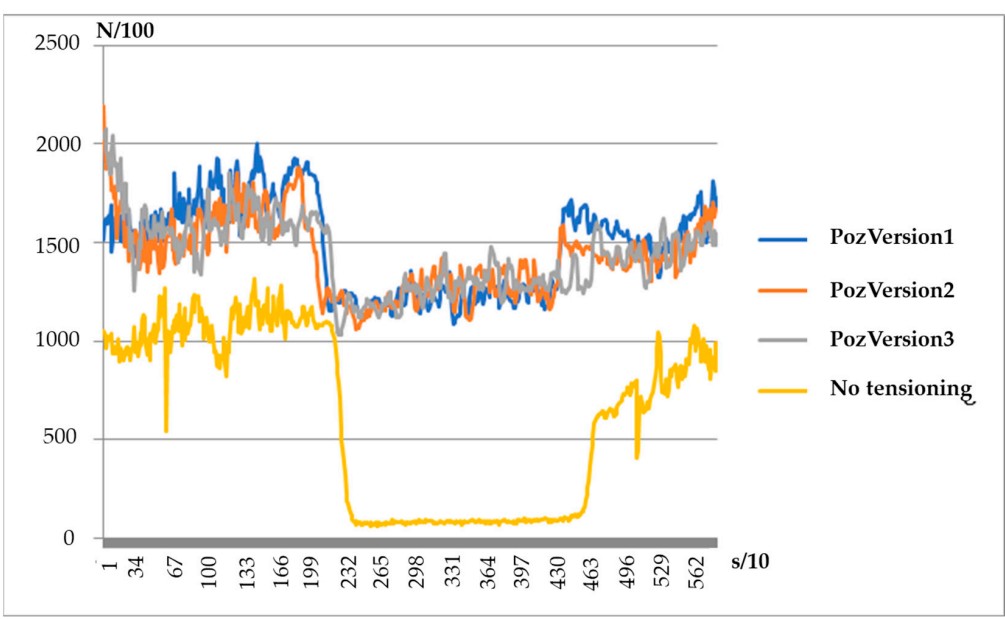

**Figure 12.** Preliminary tensioning tests.

In the previous test, the weight of the tensioning device was not considered, and it was observed that it has an influence, therefore the tension from the fiber was released and new offsets in the force measuring device were set. The tool weight was compensated very well, but we still can see a drop from 1500 to 1100 [N/100] (Figure 13), which is not so big considering the initial version of the system. However, it is not providing a very good solution for the moment.

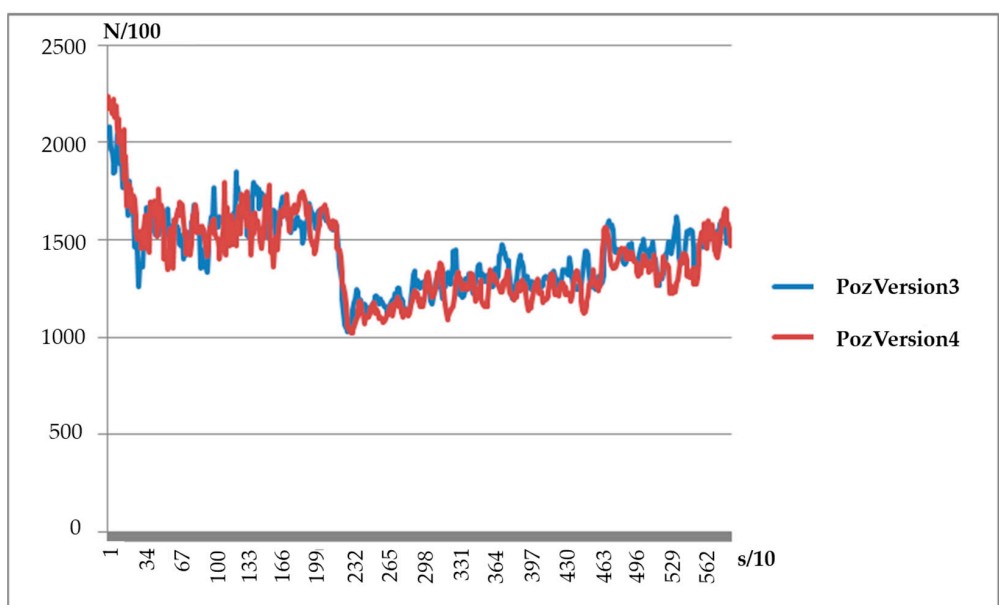

**Figure 13.** Tool weight compensation.

The automatic fiber tensioning system is partially functional only by using an amplifier in the force position conversion.

The next step was to apply a close loop controller with a regulator similar to P or PI to observe the influence on the fiber tensioning. All these tests were applied to a simple matrix. The next steps will be to apply the tensioning device together with a robotic collaborative system with high complexity matrix. The reference tension is set to 1400 [N/100].

The position formula is made of:

$$AmpFactor = \frac{MaxRotation}{MaxError} \tag{51}$$

$$ForceError = ReferenceForce - MessForce \tag{52}$$

$$PozP = AmpFactor * ForceError \tag{53}$$

For Version 1 (PozPV1) we test with the following default values:
MaxRotation = 10.
MaxError = 2000.
ReferenceForce = 1400.
For Version 2 (PozPV2) we test with the following default values:
MaxRotation = 15.
MaxError = 2000.
For Version 3 (PozPV3) we test with the following default values:
MaxRotation = 20.
MaxError = 2000.
For Version 4 (PozPV4) we test with the following default values:
MaxRotation = 10.
MaxError = 1300.
For Version 5 (PozPV5) we test with the following default values:
MaxRotation = 15.
MaxError = 1300.
The tests from Figure 14, with the proportional regulator, show that version 3 of the parameters brings the best compensation force results.

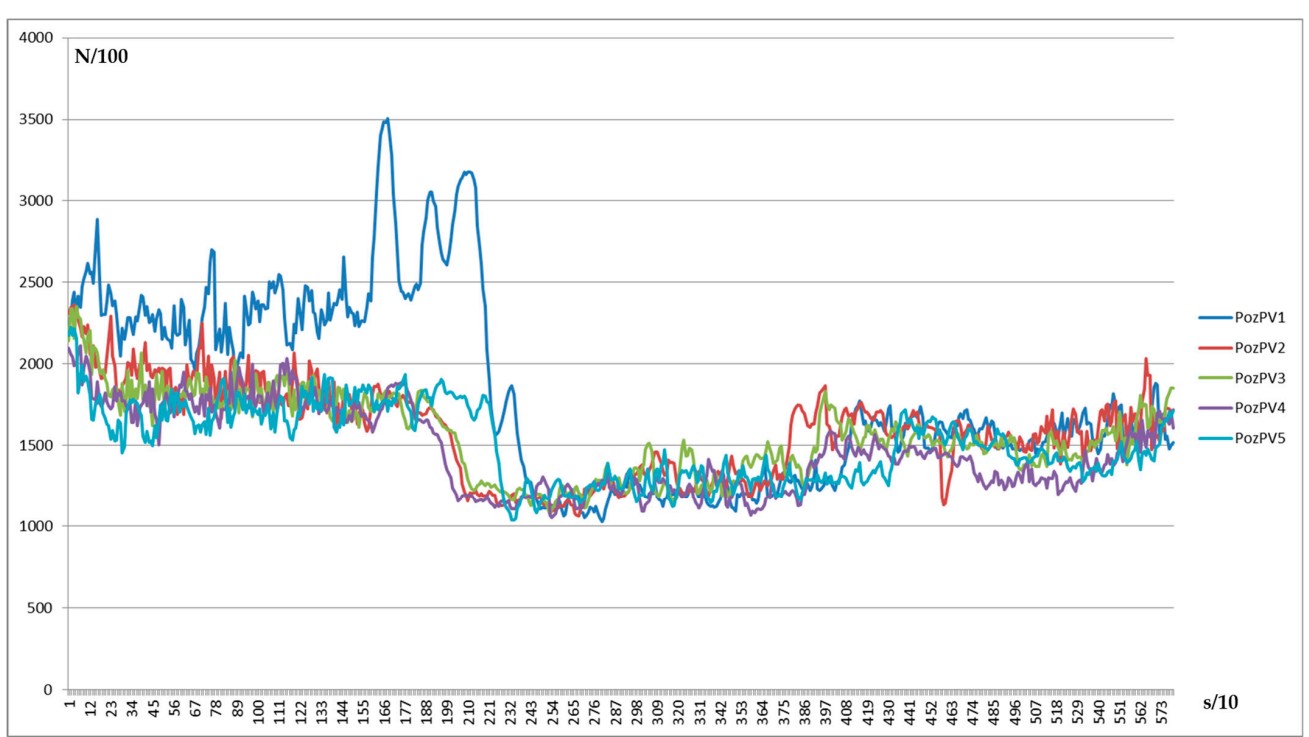

**Figure 14.** Controlled fiber tensioning results.

Even with a Proportional control, the improvements are visible (Figure 15), but not so big, so this Proportional Regulation is only the first step.

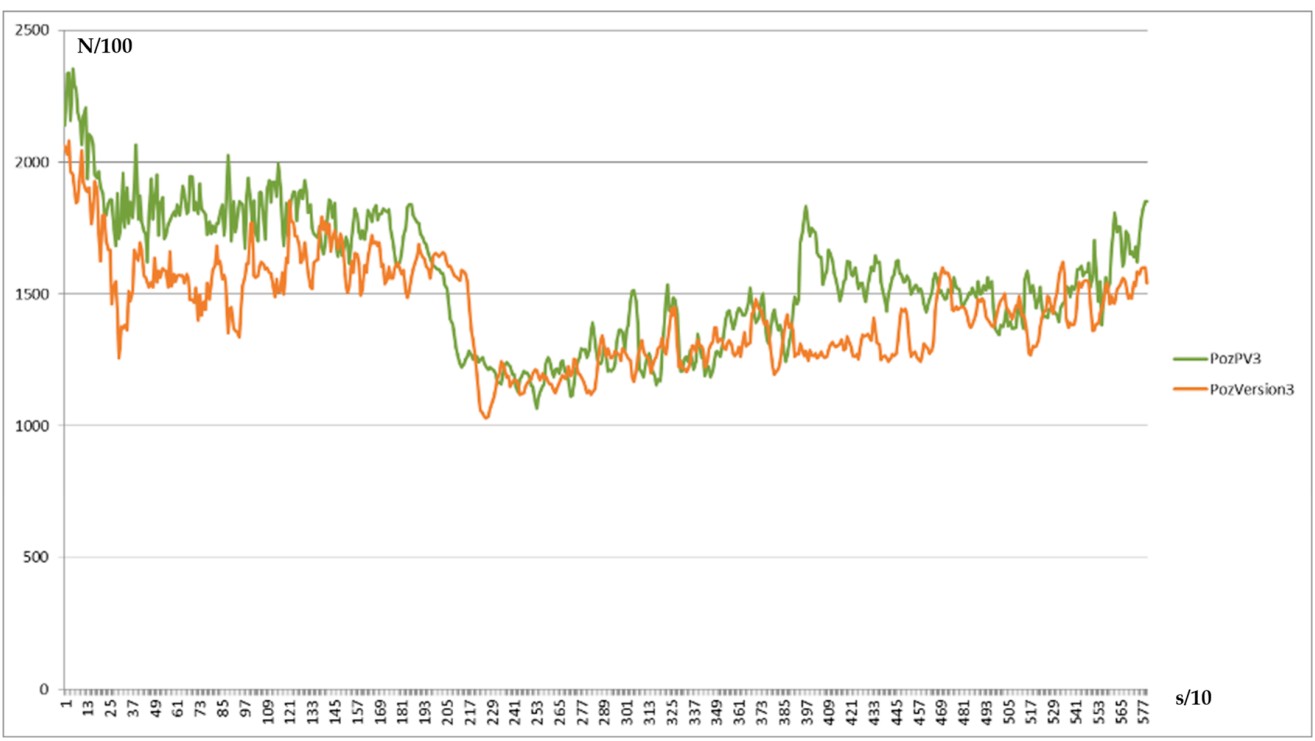

**Figure 15.** Proportional control system.

A dynamical formula coming from a bipolar combined with the Proportional control method was also tested, like:

$$\begin{cases} -100 < ForceError < 100, \; PozP = AmpFactor * ForceError; \\ Force\;error > 100, \begin{cases} Force\;error > 400, \; PozP = PozP - 0.3; \\ Force\;Error < 400, \; PozP = PozP - 0.1; \end{cases} \\ Force\;error < -100, \begin{cases} Force\;error < -400, \; PozP = PozP + 0.3; \\ Force\;Error > -400, \; PozP = PozP + 0.1. \end{cases} \end{cases} \tag{54}$$

With a bipolar version of the controller, big influences between some limits appear (Figure 16). On average, the control is better but it is not possible to consider this option because of the big fluctuations.

The last version of Proportional control shows a smaller drop but is not settling the force value on the reference, which can be observed, in the first period it is approximately 1600 (N/10). The last version of PI is stabilized at the value of 1400 but it has a bigger drop.

If we make an average of all the measured values, the result is 1402. Very close to the reference.

The problem now is the big oscillations and especially the drop at the middle of the trajectory. This is the area where the direction of the deposition head is changed from a direction opposite to the roll to a direction in the same way as the fiber source roll.

The necessary time for the motor to compensate for the change is approximately 2 s, resulting from the motor control system.

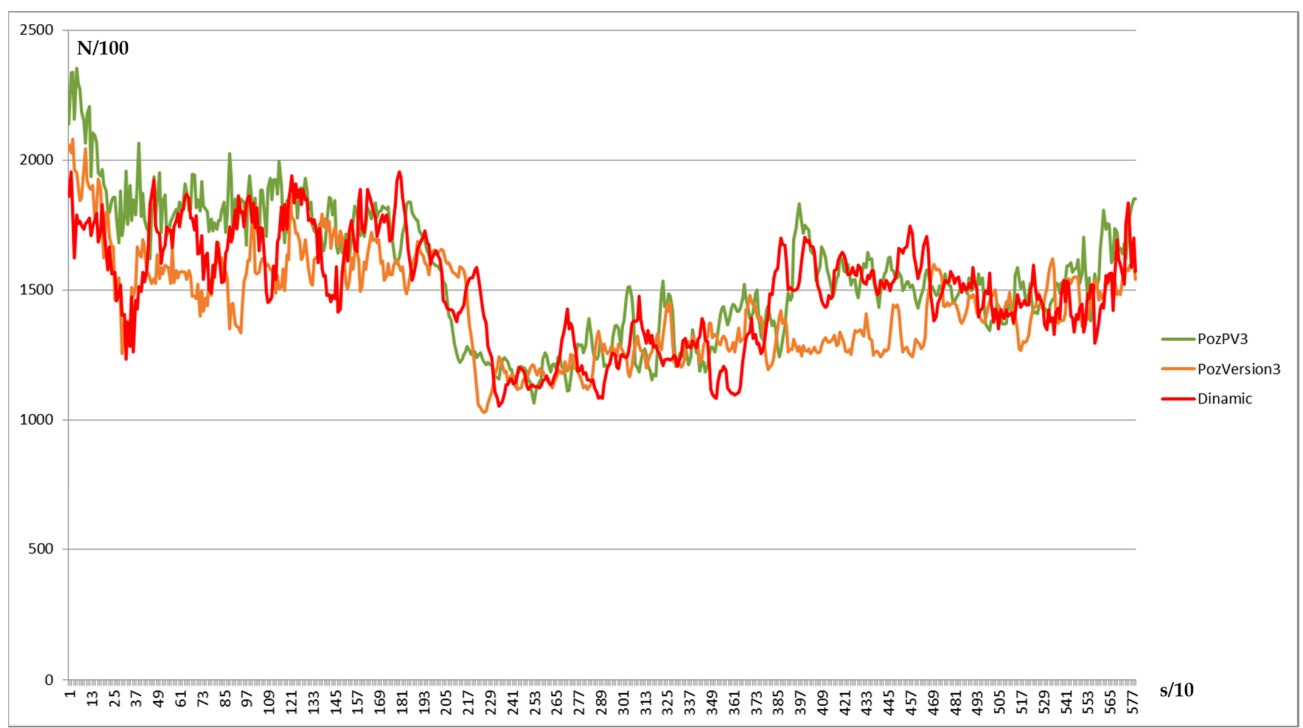

**Figure 16.** Bipolar control system.

*3.2. Control System Proposal*

Considering the results from the Proportional control, the better solution is to add an Integrator to make the settling faster. Therefore, to implement a PI control with the following general consideration:

MaxRotation = 15;
MaxError = 2000;
a1 = −1;
SamplingPeriod = 0.073;
Ti = 10;
Referenceforce = 1400;

$$AmpFactor = \frac{MaxRotation}{MaxError} \tag{55}$$

$$b1 = -1 * AmpFactor \tag{56}$$

$$b0 = AmpFactor * (1 + \frac{SamplingPeriod}{Ti}) \tag{57}$$

$$ForceError = ReferenceForce - MessForce \tag{58}$$

$$PozPIPrev_i = PozPI_{i-1} \tag{59}$$

$$ForceErrorPrev_i = ForceError_{i-1} \tag{60}$$

$$PozPI = -a1 * PozPIPrev + b0 * ForceError + b1 * ForceErrorPrev \tag{61}$$

In the first real tests, we considered the source role influence null and the roll was unrolled manually so the tensioning system had to compensate only for the robot trajectory differences. The compensation can be observed in Figure 17.

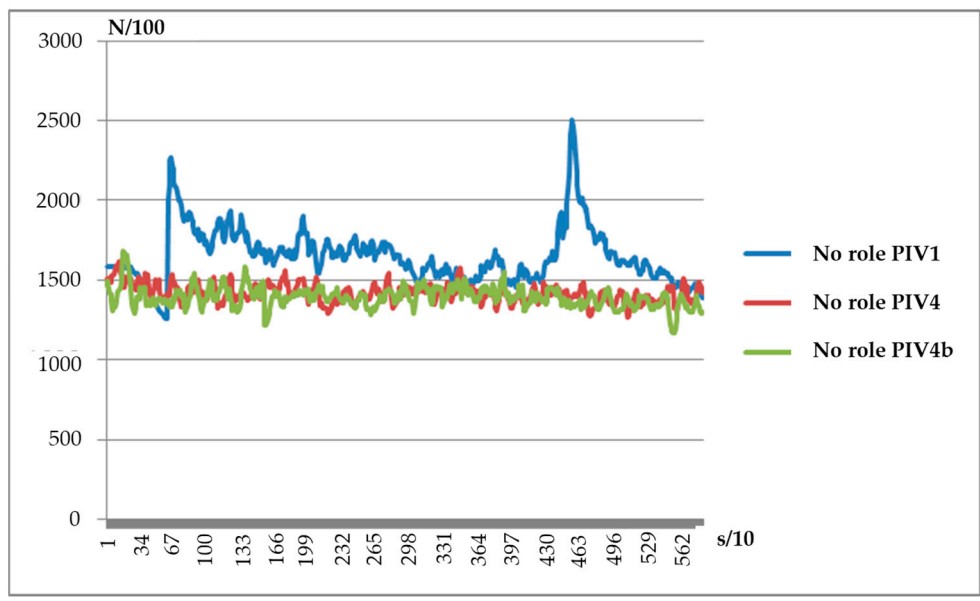

**Figure 17.** No role tension influence system measurements.

From the above figure, it is clear that applying a good formula, as described in the algorithm below, for the control system and having no perturbation, the tensioning can be made very good.

*if (comandaRot > comandaRotMaxima)*

   *{*

     *comandaRot = comandaRotMaxima;*

   *}*

*if (comandaRot < (−1 \* comandaRotMaxima))*

*{*

     *comandaRot = (−1 \* 17);*

   *}*

The problem is when the influences from the fiber source role are not null.

Some adjustments were made to the formula and the results are the following:

V1:

$$AmpFactor = \frac{MaxRotation}{MaxError} \tag{62}$$

In this version, it can still be observed the force drop in the middle of the trajectory. This means the amplifier is not big enough.

V4:

$$AmpFactor = 5 * \frac{MaxRotation}{MaxError} \tag{63}$$

Here can be observed a correction on the dropping, but it is still there. This is because the correction is made faster because of the amplifier.

V6

$$AmpFactor = 10 * \frac{MaxRotation}{MaxError} \tag{64}$$

Having the result from the previous test, we consider that an even bigger amplifier will solve the problem. From the measured results, it can be seen the dropping value decreasing and the average value is very close to the reference force set in the parameters (Figure 18).

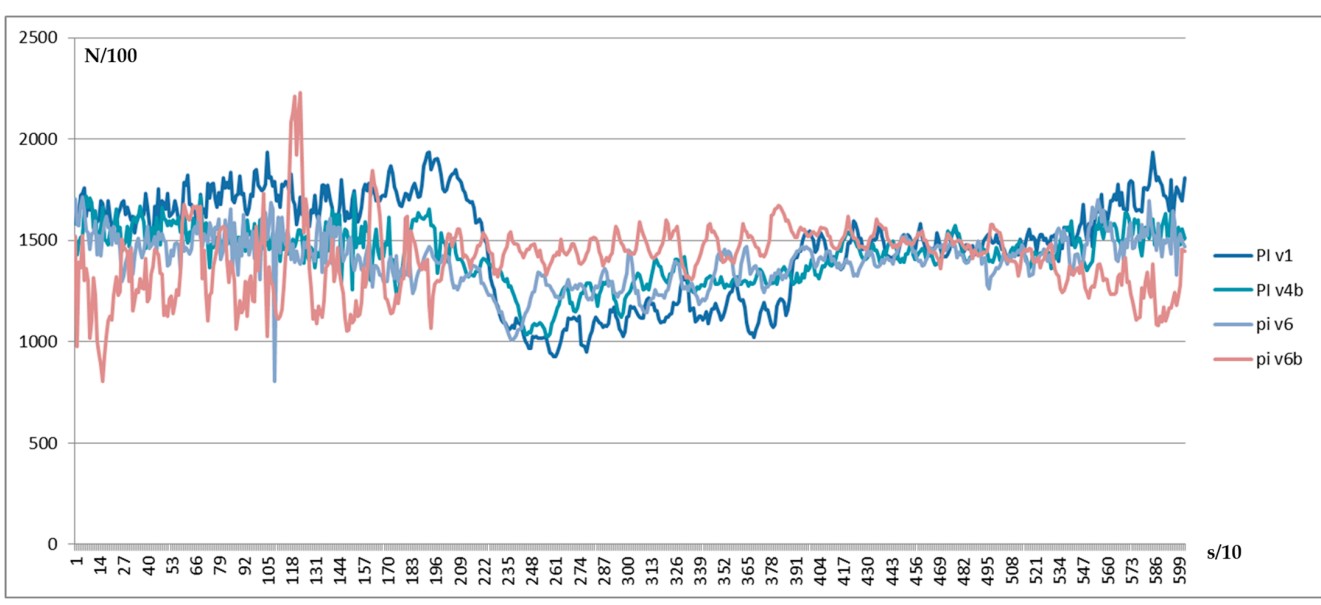

**Figure 18.** Limited control system.

## 4. Results and Analyze

To the previous test were added limits because of the hardware implication of the tensioning device, but also to have better control of the positioning. Because of the limits, big oscillations of the force can appear if the process is not kept under control for the complete time (Figure 19).

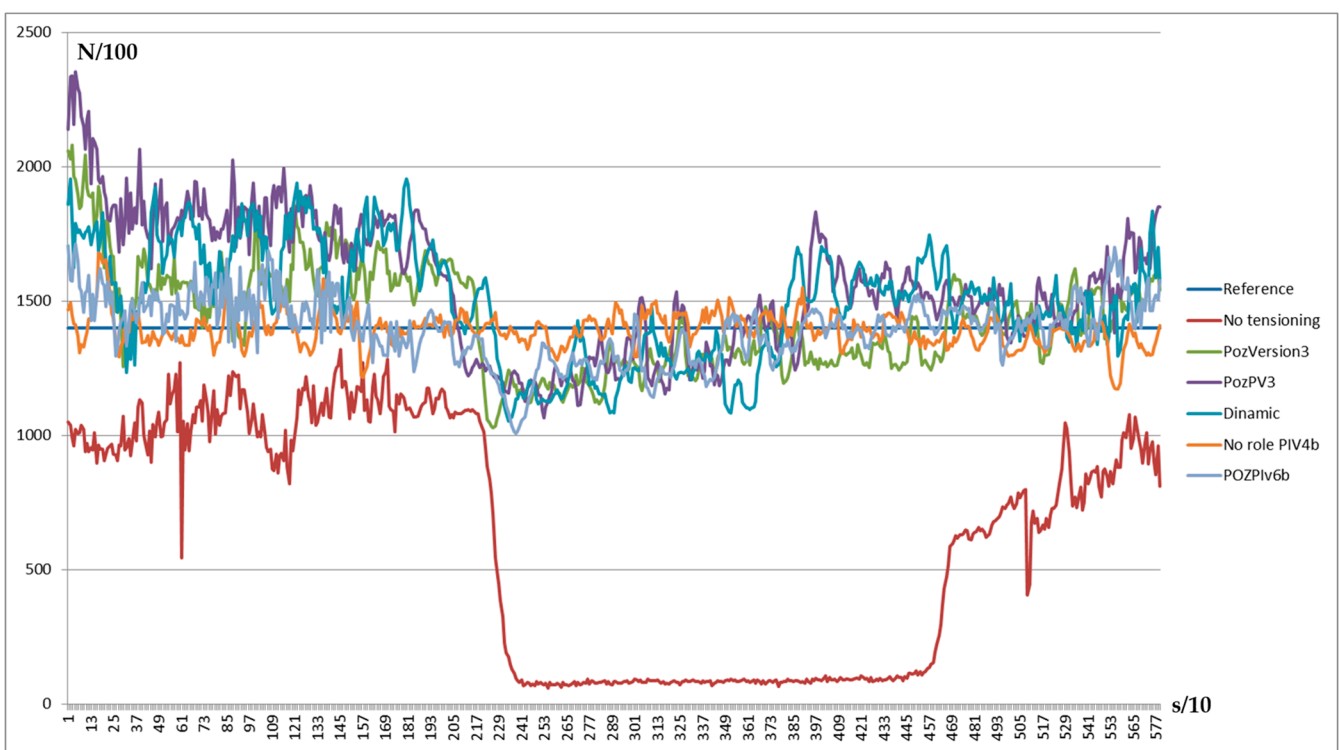

**Figure 19.** An increased amplifier and limited values control system.

As can be observed, if a big amplifier with limits is used, but also taking into consideration the other constants that a PI controller brings, better results can be obtained compared with the previous tests with a P controller where it is only about the amplifier.

As we mentioned before, all of these tests were made on a matrix with a medium complexity level. With this was possible to observe the influence of the fiber source role in all directions and to compensate for the influence (Figure 20).

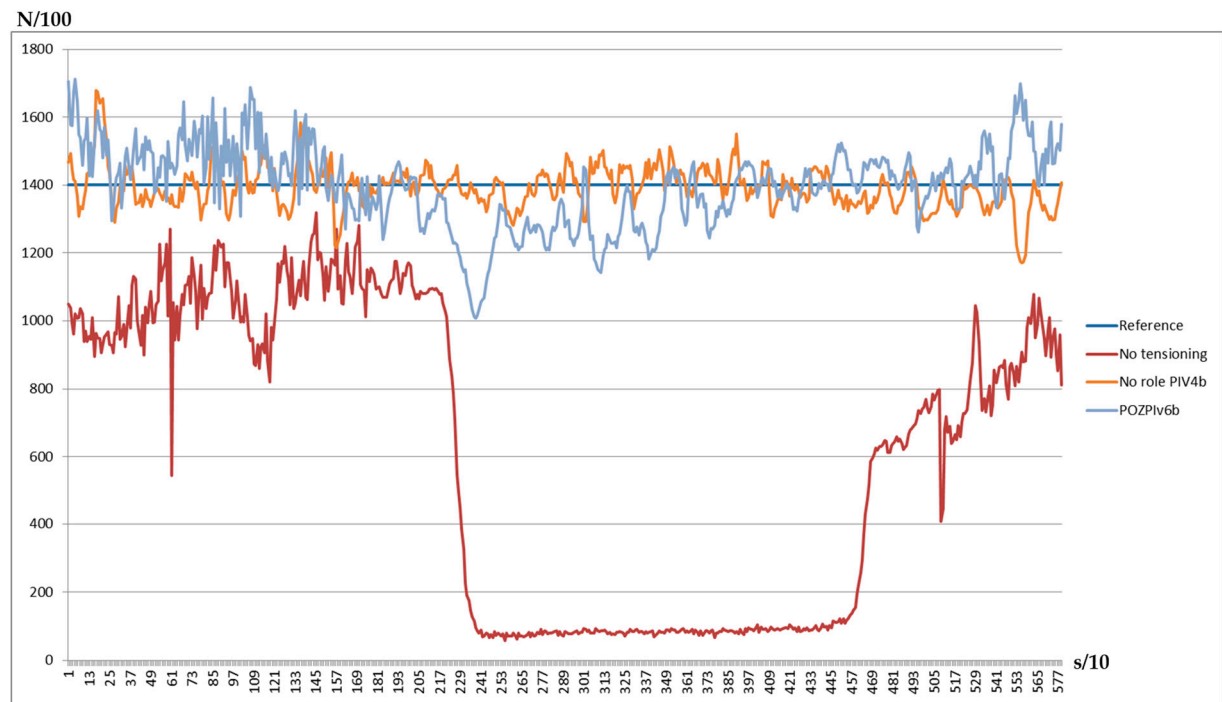

**Figure 20.** General control system results.

The previous steps and the same tensioning software structure can be used on a complex matrix as well. First of all, the trajectory of the robots and the collaboration model were created virtually using ABB RobotStudio 5.15.02 software. In the simulation, all necessary tools were imported and the speed and the trajectory reachability were tested (Figure 21).

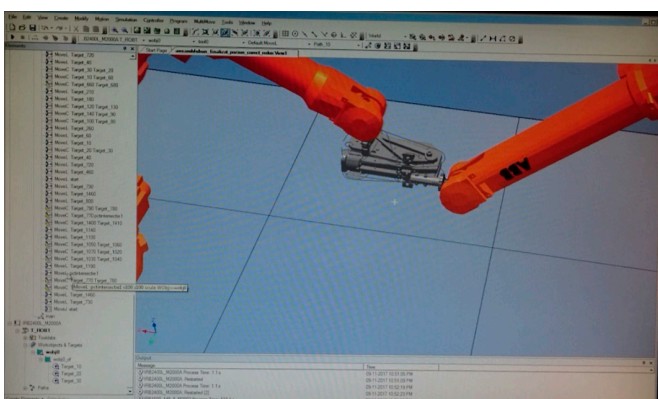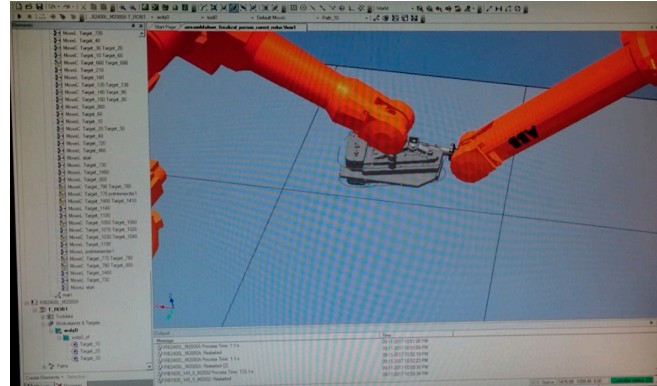

**Figure 21.** Robot studio robot collaboration simulation.

The trajectory identified in the simulation was implemented in the real system and the collaboration instruments were used via the master platform, which was making the connection between the robots and gives the signals between them (Figure 22).

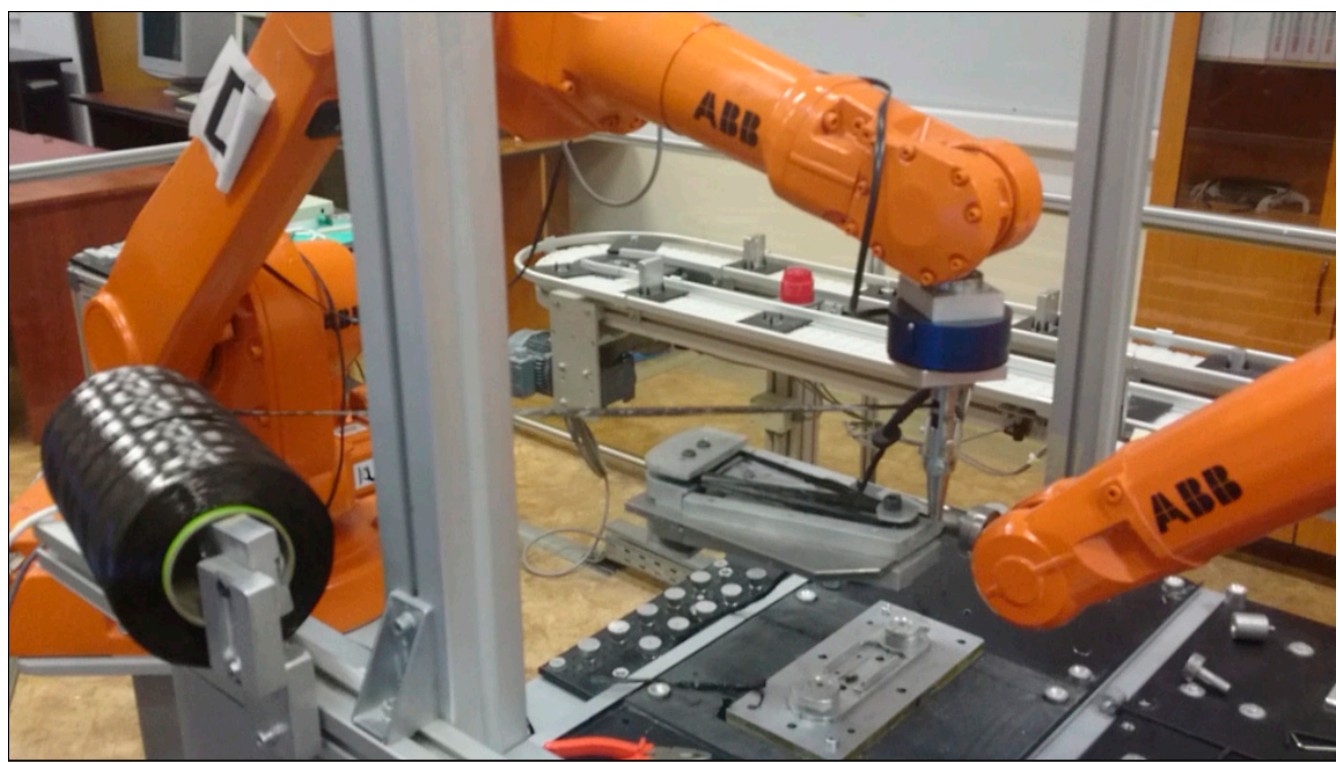

**Figure 22.** Real robot collaboration system.

The collaboration process realizes the winding on both sides of the winding die (Figure 23). The winding die has to be rotated on some points to be able to wind on both sides. There were implemented signals for turn confirmation or position reach-ready to turn.

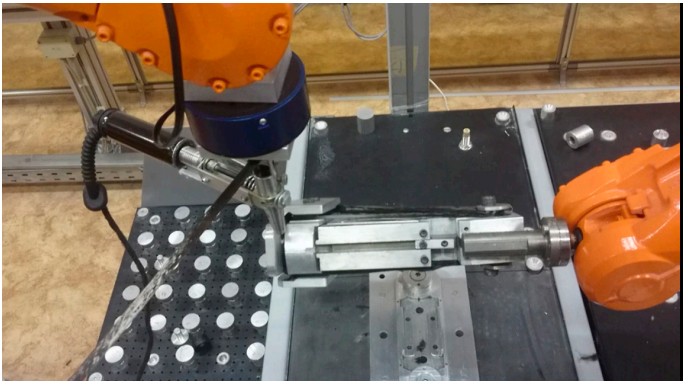
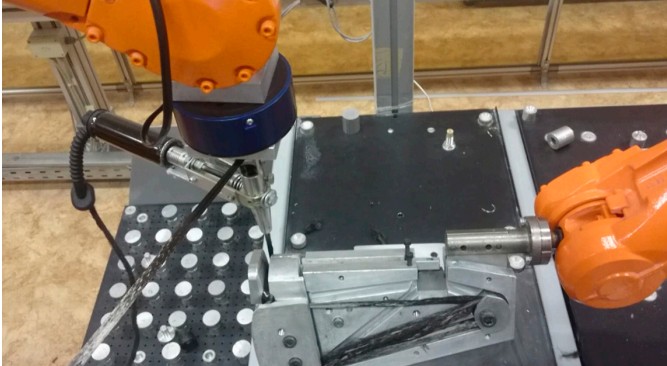

**Figure 23.** Winding dies rotation process.

The advantage of this tensioning device is that, even if it is mounted on a single robot, it is active also when the other robot is creating tension, due to the rotation, meaning the tension is constant including in the rotation moments.

In Figure 24, the measured force without the role influence and the tensioning device can be seen and the result with both in action, the perturbation, and the tensioning device with a PI regulation implemented. Having the results from the previous implementation, improvements can be made from the mechanical implementation and trajectory corrections. In Figure 25, a trajectory path on a single plan is presented.

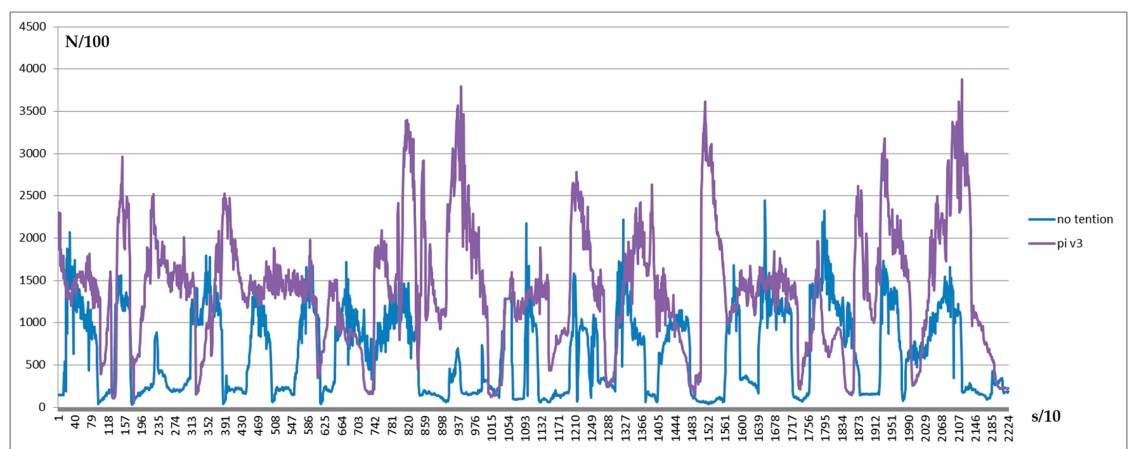

**Figure 24.** Complex winding dies force measurements.

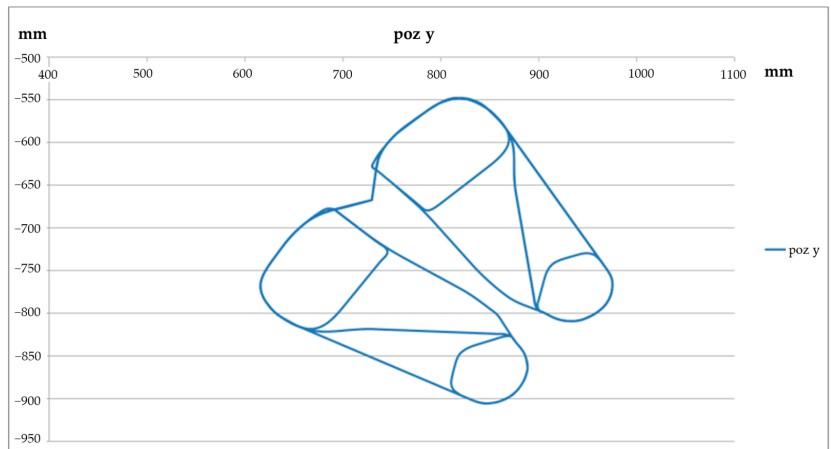

**Figure 25.** Complex trajectory plan.

According to the model, the fiber is tensioned between the matrix and the winding head. If the trajectory is similar to the one presented in Figure 26, then the tension is loos for a short moment because of the trajectory, but this is recovered after 2–3 measuring points if we apply the control system with a PI formula, as it can be observed in Figure 27.

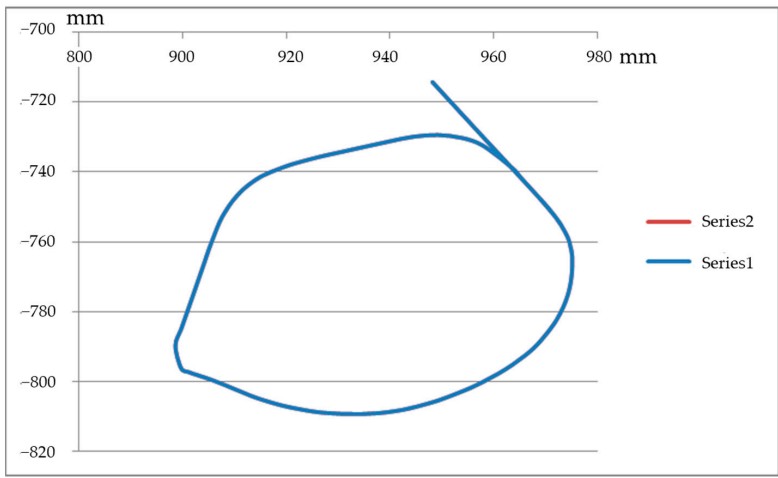

**Figure 26.** Trajectory sample.

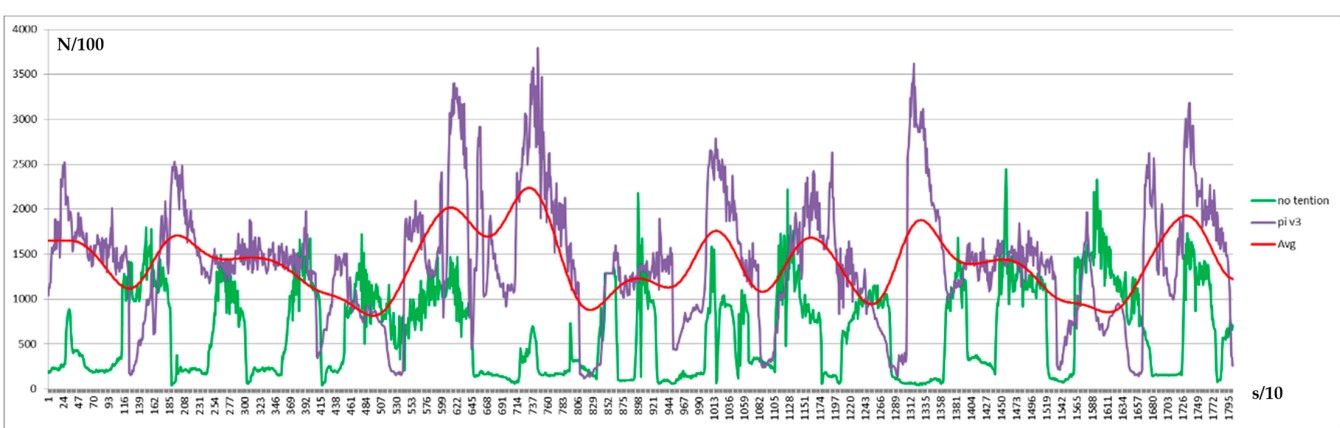

**Figure 27.** Complex matrix tensioning control results.

## 5. Conclusions

The experimental tests carried out with one robot, were made to verify and prove the fiber winding methods and models. In the first phase, to determine the nominal values of the stress in the fiber, experimental tests were made to analyze the influences of the frictional forces from the resin supply and impregnation subsystem, on the tension winding process. As a result of these tests, in addition to the fact that the length of the wound fiber decreases with the increase in the value of the tensioning effort, the influence of the resistances in the feeding and impregnation subsystems on the fiber effort generated through the tensioning device integrated into the winding head was highlighted.

From the analysis of the resulting graphs, it is observed that the best results are obtained by implementing the Proportional-Integrator type regulator, which is based on the amplification factor.

The experimental tests performed with two robots were the basis for identifying the nominal parameters of the winding process. After repeating the carbon fiber winding process, under identical conditions, but with different settings of the tension control systems, the best results are obtained with the improved Proportional-Integrator regulator, which ensures the smallest deviation throughout the trajectory, $\pm 0.5$ N; the average value of the measured effort being 14.75 N, very close to the proposed target, 15 N.

Starting from the results obtained from the implementation of the carbon fiber tensioning with force adjustment algorithm, for the winding trajectory with a single pass through each point, the trajectory was later extended for the entire length of the deposition path (in multiple layers) of the carbon fiber in the mold, using the same algorithm, with very good results on the entire trajectory.

The implemented carbon fiber tensioning system has the advantage that it can influence the carbon fiber tensioning even during the mold rotation process. The integrated system with collaborative robots can thus be used, together with the carbon fiber tensioning sub-system, supervised by the master controller, without the intervention of the human factor, to ensure a winding that leads to the achievement of the performances im-posed on the final product, defined by the designer.

Following the experimental tests carried out and the results obtained, it is shown that the winding process with two robots and a tensioning system can realize fiber windings with variable tension in molds with complex configurations (3D) of the winding paths to obtain compact structures of the constantly tensioned fiber bundle, according to product requirements.

In summary, within this research, two groups of findings were outlined: the first, concerns the planning, simulation, generation, and implementation of the trajectories of robots to work collaboratively; the second group, is related to the controlled tensioning of carbon fiber using a new tensioning device with a control and command subsystem based on the mathematical winding model.

This winding system with collaborative robots proposed by this research has the potential to step out from the development phase and to be introduced as a standard system for carbon fiber product developers. This is therefore addressed to the large manufacturers of structures made of composite materials who want both a large series production without interruptions and with minimal human influence, but also to reduce the time required for changeovers to a minimum, having prepared the model of the new mold, the tensioning adjustment is automatically done. The limitations of the system can only be seen from the perspective of the need to continue planning and generating the robots' trajectory in the virtual environment before implementation on the robots, a time-consuming action and with the necessity of highly qualified human resources. However, this limitation appears only when implementing a new part type.

**Author Contributions:** Conceptualization, M.P.S. and G.L.M.; methodology, M.P.S.; software, M.P.S.; validation, G.L.M. and R.G.B.; formal analysis, M.P.S.; investigation, M.P.S.; resources, G.L.M.; data curation, M.P.S.; writing—original draft preparation, M.P.S.; writing—review and editing, R.G.B.; visualization, R.G.B.; supervision, G.L.M. All authors have read and agreed to the published version of the manuscript.

**Funding:** This research received no external funding.

**Data Availability Statement:** Not applicable.

**Conflicts of Interest:** The authors declare no conflict of interest.

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
