# Peer review of "Fiber Tensioning Systems in a Robotized Winding Procedure for Composite Materials Building Processes"

_electronics, doi:10.3390/electronics12020254_

Round 1

Reviewer 1 Report

The paper “Fiber tensioning systems in robotize winding procedure for 2 composite materials building processes” by Sbanca et al introduces a new complete carbon fiber system based on winding process for high volume of parts and unique objects production. In general, this article needs significant further efforts on presenting the novelty, evidence, and necessity of this invention. The abstract failed to capture the essence of this article. Audience from different engineering background could not understand the concept and result of the new developed system. The introduction and literature review should be combined into a much more concise format as this is an article instead of review article. The authors should point out what is the limit of current technology and the proposed solution plan with supporting references. The extensive literature review session will confuse the audience. The authors also need extensive efforts on the wording of manuscript as current version is very elusive. Also, the authors should reduce the number of figures in the articles. Only core figures is necessary to present in manuscript as other redundant information can be included in supplements.     

Author Response

Thank you for all your inputs. It is very helpful for us. We reviewed the manuscript and attach you can also find our inputs point by point.

Reviewer 2 Report

The paper ‘Fiber tensioning systems in robotize winding procedure for composite materials building processes’ shows the possibility to conduct the winding  process of carbon fiber using two industrial robots, one to realize the winding and the other one to hold and rotate the mandrel.

The theme of paper is interesting, but it seems at a preliminary stage, the form is not appropriate for a scientific journal  and several aspects are not opportunely clarified and presented. For a successive r-submission the following comments needs to be tackled an answer:

- In Fig. 5 is shown the one loop control for controlling the tensioning itself using the force feedback and you claim that the control system is adjusted manually (parameters of PID). But there are some applications of rolling systems controlled as in Fig. 5 where auto-adaptative techniques are used for tuning the controller parameters automatically; please consider and discuss these studies also in comparison with your proposal;

- In Eq. (1) could you please explain better why you consider the first-order dynamic (also a second-order could match the behavior in Fig.7) ? It is a very important assumption for the following simulations, so it should be justified;

- The simplification in Eq. (8-11) is not clear and, probably, it is not necessary;

- Figs. 11-13 should be modified with white background and adding captions in a more scientific format;

-Figs 14-15 needs to be improved;

- Fig. 21 lacks of indication of units (x and y axis)

- Please explain the terms in Eq 45-47;

- Eg. (49) is unclear;

- Fig. 22 misses the indication of axis units;

- Eq. (50) unclear;

- Fig. 23-30 misses the indication of axis units;

- Fig. 35-36 misses the indication of axis units and are unclear;

Author Response

(The authors gave the same response as above.)

Reviewer 3 Report

This paper presents analysis of the possibility to conduct the winding using two industrial robots, one to realize the winding and the other one to hold and rotate the mandrel. Overall, the research focus of this research should be further refined.

1, The title of this paper mentioned tensioning system for continuous manufacture composite materials. However, the introduction part seems irrelated to this focus. For example, the first paragraph of this part is talk about energy. The review suggests deleting this paragraph.

2, In Line 108, “this method” refers to what? The review suggests deleting this sentence.

3, There are so many figures, However, some of them are so common, it does not make any differences to help improve the novelty of this manuscript. For example, Fig. 8, 9,10, 19, 20, 31.

4, The literature part should focus on passive or active tension control. Otherwise, please delete this part.

5, How many layers of composite materials or integrated circuit can be fabricated for your proposed method. Did authors consider registration alignment?

Author Response

(The authors gave the same response as above.)

Round 2

Reviewer 1 Report

Authors have significantly addressed my previous concerns and made extensive changes to the original documents. In general, I feel this manuscript is in good shape but still require a few more refinement.

1. The 3-page introduction is still too long from my opinion as some background may not be relevant for detailed illustrations. For example, the raw type of materials for composites, procedures for manufacturing carbon fibers, different winding processes and their pros and cons, all can be further cut or deleted. 

2. The author should do a better job on illustrating the main contributions of this article in introduction. As the part from line 175-200, the description remains elusive. I would suggest a simple summary of what is the current issue (automatization, winding process, composite materials, lack of systematic building method, etc.) followed by what is being done in this paper (new modeling method, novel system, production method). This would make everything clear and concise.

3. The abstract is very important for an article. Line 14-17 remain elusive in English expression. Separate the sentence into several sentences would be better for expression.

4. Line 10, change "analyzes" to "analyze" for grammar correction.

Author Response

Thank you very much for your opinion. Please find attach our response. 

Reviewer 2 Report

Ok, you succesfully tackled all the comments.

Reviewer 3 Report

This article analyzes the possibility to conduct the carbon fiber winding using two industrial robots connected as master slave, one to realize the winding and the other one to hold and rotate the mandrel, using an own design of an automatic fiber tensioning tool. This is an interesting topic in R2R field. Following issues should be concerned before recommending publication.

1, The organization of this manuscript should be further refine, authors should focus on tensioning system and force control. Thus, Fig. 1 , 2 and 3 cannot well support the research focus. Please delete. 

2, This is a research journal paper, not a thesis. Thus, the focus of this manuscript should be focus on tension modeling and control. The review about tensioning devices, tensioning advantages should be deleted.

3, Dis these figures from Fig. 21 to Fig. 31 can well support the research results?

4, Format of the references should be further revised.

Author Response

Thank you very much for your suggestions. Please find attached a detailed response

Round 3

Reviewer 3 Report

The authors well repond concerns. This manuscript can be considered for publication in present form.

Author Response

Thank you for your opinion. We tried to fix open points.